# EPIC-KITCHENS VISOR Benchmark
# VIdeo Segmentations and Object Relations

**Ahmad Darkhalil**[★♣]  **Dandan Shan**[★♠]  **Bin Zhu**[★♣]  **Jian Ma**[★♣]
**Amlan Kar**[♦]  **Richard Higgins**[♠]  **Sanja Fidler**[♦]  **David Fouhey**[♠]  **Dima Damen**[♣]

[♣]Uni. of Bristol, UK  [♠]Uni. of Michigan, US  [♦]Uni. of Toronto, CA  [★]: Co-First Authors

## Abstract

We introduce VISOR, a new dataset of pixel annotations and a benchmark suite for segmenting hands and active objects in egocentric video. VISOR annotates videos from EPIC-KITCHENS, which comes with a new set of challenges not encountered in current video segmentation datasets. Specifically, we need to ensure both short- and long-term consistency of pixel-level annotations as objects undergo transformative interactions, e.g. an onion is peeled, diced and cooked - where we aim to obtain accurate pixel-level annotations of the peel, onion pieces, chopping board, knife, pan, as well as the acting hands. VISOR introduces an annotation pipeline, AI-powered in parts, for scalability and quality. In total, we publicly release 272K manual semantic masks of 257 object classes, 9.9M interpolated dense masks, 67K hand-object relations, covering 36 hours of 179 untrimmed videos. Along with the annotations, we introduce three challenges in video object segmentation, interaction understanding and long-term reasoning.

For data, code and leaderboards: `http://epic-kitchens.github.io/VISOR`

## 1  Introduction

Consider a video capturing the tedious process of preparing bread, from obtaining, measuring and mixing ingredients to kneading and shaping dough. Despite being a routine task, the discrete nature of computer vision models, trained mostly from images, expects to recognise objects as either *flour* or *dough*. Capturing the transformation through pixel-level annotations has not been attempted to date. Building on top of EPIC-KITCHENS egocentric videos [12], VISOR utilises action labels and provides sparse segmentations, of hands and active objects, with a rate of annotations so as to represent both short (e.g. 'add salt') and long ('knead dough') temporal actions. In Fig. 1, we introduce sample annotations from one video of VISOR.

Rich pixel-level annotations have transformed image understanding [17, 23] and autonomous driving [10, 13], amongst other tasks. By augmenting action labels with semantic segmentations, we hope to enable complex video understanding capabilities. Previous efforts to incorporate spatio-temporal annotations and object relations [15, 16, 19, 22] have only featured bounding boxes. A few seminal video datasets [31, 41, 43] provide pixel labels over time, but are often short-term, and are not associated with fine-grained action labels. We compare VISOR to other datasets in §2.

We obtain VISOR annotations via a new pipeline consisting of three stages that we describe in §3: (i) identifying *active objects* that are of relevance to the current action, as well as their semantic label; (ii) annotating pixel-level segments with an AI-powered interactive interface iterated with manual quality-control checks; and (iii) relating objects spatially and temporally for short-term consistency and hand-object relations.

36th Conference on Neural Information Processing Systems (NeurIPS 2022) Track on Datasets and Benchmarks.

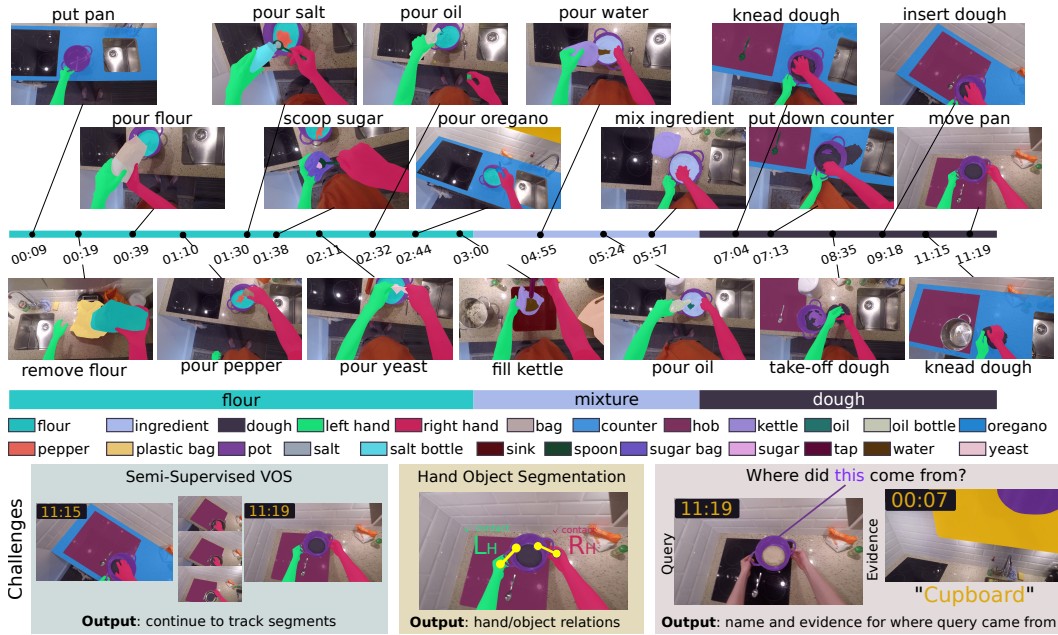

Figure 1: **VISOR Annotations and Benchmarks.** Sparse segmentations from one video (P06_03), where flour turns into dough over the course of over 11 minutes. The colours of the timeline represent the stages: flour→mixture→dough. By building off of the EPIC-KITCHENS dataset, our segments join a rich set of action labels. We use the annotations to define three challenges at different timescales: (i) Semi-Supervised VOS over several actions; (ii) Hand↔Object Segmentation and contact relations in the moment; and (iii) *where did this come from?* for long-term reasoning.

In total, we obtain annotations of 271.6K masks across 50.7K images of 36 hours, interpolated to 9.9M dense masks, which we analyse in §4. Moreover, these new annotations directly plug in with a strong existing ecosystem of annotations, from action labels to closed-vocabulary of objects.

In §5, we use these new annotations to define three benchmarks, depicted in Fig. 1 (bottom). Our first task, *Semi-Supervised Video Object Segmentation (VOS)*, requires tracking object segments across *consecutive actions*. We extend previous VOS efforts [31, 41, 43] by tackling 2.5-4x longer sequences. Our second task, *Hand Object Segmentation (HOS)*, involves predicting the contact between hands and the active objects, and extends past efforts by predict both the relation and accurate segmentations of hands and objects. Our final task, *Where Did This Come From (WDTCF)?*, is a taster benchmark for long-term perception with scene segmentation, where one traces back a given segment to the container it came from (e.g., milk from the fridge, and a plate from a particular cupboard) and identifies the frame(s) containing evidence of the query object emerging from the container.

## 2 Related Efforts

We compare VISOR with previous efforts that have *both* pixel-level and action annotations in Table 1. We divide our comparison into three sections: *basic statistics* on the videos themselves, *pixel-level annotation statistics* on images annotated, masks and average masks per image and *action annotations* where entities or action classes have been labelled. Our videos are significantly longer. On average, VISOR videos are 12 *minutes* long. To make the challenge feasible to current state-of-the-art models, we divide our videos into shorter sequences that are 12 *seconds* long on average. VISOR is also the only dataset that combines pixel-level segmentations with both action and entity class labels.

The closest effort to VISOR is that of the seminal UVO work [41] that annotates videos from the Kinetics dataset [7]. As every video in Kinetics captures a single action, UVO only offers short-term object segmentations, focusing on translated objects and occlusions. In contrast, VISOR annotates videos from EPIC-KITCHENS [11] that capture sequences of 100s of fine-grained actions. VISOR segmentations thus capture long-term object segmentations of the same instance undergoing a series of transformations – the same potato is seen being picked from the fridge, washed, peeled, cut,

Table 1: **Comparison with Current Data.** Compared to past efforts featuring both pixel- and action-level annotations, VISOR features longer sequences and provides the largest number of manually annotated masks on diverse actions and objects. *:Stats from train/val public annotations. $^\dagger$Avg VISOR video is **12 *minutes*** long. We divide these into sub-sequences w/ consistent entities (See §3). Masks are semantically consistent, through class knowledge, across all videos.

| | Basic Statistics | | | Pixel-Level Annotations | | | Action Annotations | | |
| Dataset | Year | Total Mins | Avg Seq Ln | Total Masks | Total Images | Avg Masks per Image | Actions | #Action Classes | #Entity Classes |
|---|---|---|---|---|---|---|---|---|---|
| EgoHand [3] | 2015 | 72 | - | 15.1K | 4.8K | 3.2 | - | - | 2 |
| DAVIS [6] | 2019 | 8 | 3s | 32.0K | 10.7K | 3.0 | - | - | - |
| YTVOS [43] | 2018 | 335 | 5s | 197.2K | **120.4K** | 1.6 | - | - | 94 |
| UVOv0.5 (Sparse) [41] | 2021 | 511 | 3s | *200.6K | 30.5K | **\*8.8** | 10,213 | 300 | - |
| VISOR (Ours) | 2022 | **2,180** | **12s**$^\dagger$ | **271.6K** | 50.7K | 5.3 | **27,961** | **2,594** | **257** |

and cooked. Object transformations are also significantly more varied, beyond translation and deformation, offering a challenge previously unexplored. Ensuring long-term temporal consistency in VISOR requires a more complex framework for relating objects over sequences and through semantic classes. In doing so, we take inspiration from UVO [41] in how sparse segmentations were used with bi-directional interpolations to obtain dense annotations (See appendix 3.4).

VISOR is also related to efforts that capture hand-object relations, including for identifying active objects [35, 4], next active objects [33, 25], object articulation [44] and grasping [18, 39, 38, 40]. Distinct from these works, we provide the first pipeline, annotations and benchmark suite for pixel-level segmentations.

## 3 VISOR annotation pipeline

Our annotation pipeline consists of multiple stages. In the first stage, we identify the frames and *active* entities to be annotated (§3.1). Having identified *what* should be segmented, we obtain pixel-level annotations, substantially accelerated by an AI-powered tool. To achieve consistency, we subject annotators to extensive training, and employ manual verification (§3.2). Finally, we collect additional annotations that are needed for our challenges (§3.3). We next describe our pipeline in detail.

**Background: EPIC-KITCHENS Dataset and Annotations.** We build on the large-scale egocentric EPIC-KITCHENS-100 dataset [11], collected with University of Bristol faculty ethics approval and signed consent forms of participants (anonymous). Participants wear a head-mounted camera, start recording before they enter the kitchen and only stop recording when they leave the room. These videos have fine-grained action labels comprising: (i) video file (e.g., P06_108); (ii) start and end times of an action (e.g., from 03:52.6 to 03:58.4); (iii) short open-vocabulary narration describing the action (e.g., 'take mozzarella out of packet'); (iv) closed-vocabulary of *verb* and *noun* classes (e.g. mapping 'mozzarella' to the class 'cheese'). Table 2 of [11] compares EPIC-KITCHENS to publicly available action recognition datasets in size, annotations and classes (e.g., [7, 16, 37]).

### 3.1 Entities, frames and sub-sequences

Our first stage aims to identify what *data* and *entities* ought to be annotated. Our data selection aims to find clear frames to annotate and distribute annotation uniformly across actions rather than time, to cover actions of varying length. Entity selection aims to find all active objects, relevant to the action, including tools, crockery and appliances. This step is needed since the narrations do not include objects that are implied by common-sense, and are different per sample: e.g., one example of the action 'peel carrot' might involve a peeler and a bowl while another might use a knife and chopping board. Note that as videos are captured in people's homes, frames can be highly cluttered with background/irrelevant objects we do not wish to segment.

We show the pipeline for the selection of frames and entities in Fig. 2. We divide the untrimmed videos into sequences comprising 3 non-overlapping actions. This achieves a mean of 12 seconds of temporally consistent annotations. We chose 3 actions so as to introduce slight but reasonable challenge to previous video segmentation datasets, increasing sequence lengths from 3-5s to 12s (see Table 1). We annotate 6 frames per sequence, avoiding blurred frames when possible for better object boundaries. We candidate active entities to annotate from parsing nouns in narrations, correlations with actions from parsed verbs (e.g., 'cut' and 'knife') and manual annotations. Annotators select the

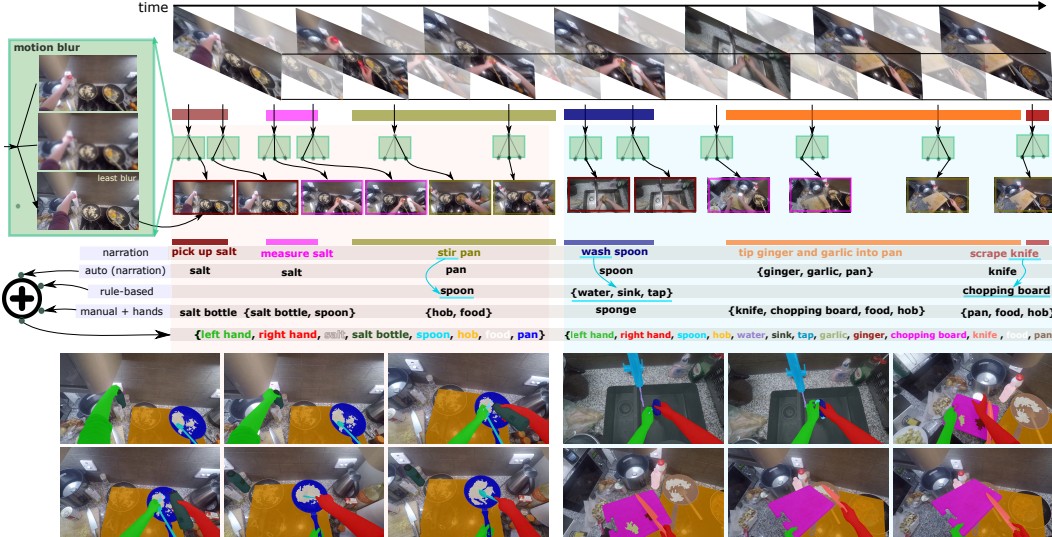

Figure 2: **Part of the VISOR Annotation Pipeline.** We divide untrimmed videos to subsequences of 3 consecutive actions, sampling 6 frames/seq. We find entities to annotate via the union of automatic parsing of narrations, manual rules, and human annotations. These are then manually segmented.

Table 2: **VISOR splits statistics**. Details of VISOR splits.

| VISOR splits | Train | Val | Train+Val | Test | Total |
|---|---|---|---|---|---|
| # Kitchens | 33 | 24 (5 unseen) | 38 | 13 (4 unseen) | 42 |
| # Untrimmed Videos | 115 | 43 | 158 | 21 | 179 |
| # Images (and %) | 32,857 (64.8%) | 7,747 (15.3%) | 40,604 (80.0%) | 10,125 (20.0%) | 50,729 |
| # Masks (and %) | 174,426 (64.2%) | 41,559 (15.3%) | 215,985 (79.5%) | 55,599 (20.5%) | 271,584 |
| # Entity Classes | 242 | 182 (9 zero-shot) | 251 | 160 (6 zero-shot) | 257 |

entities that are present per frame from this over-complete list. A showcase of the dataset's diversity in actions and entities is visualised in Fig. 3 and more details are in the appendix.

## 3.2 Tooling, annotation rules and annotator training

During the project's 22 months, we worked closely with 13 annotators: training, answering questions, and revising their annotations. A more complete description of the annotation pipeline and interface, including cross-annotator agreement, is included in the appendix.

**AI-Powered Segmentation Tooling.** To reduce the cost of pixel-level annotation, we use TORonto Annotation Suite (TORAS) [20], which uses interactive tools built from variants of [24, 42]. The interface combines a mix of manual and AI-enhanced tools that we found to dramatically improve the speed of obtaining accurate annotations. The AI models were trained on generic segmentation datasets like ADE-20K [45]. We found they generalised well to our domain as shown in [2, 8, 24].

**Annotator Recruiting, Onboarding, and Rules.** We hired freelance annotators via Upwork. They received an overview of the project, training sessions with the tool, and a glossary of objects. We devised rules to facilitate consistent segmentations across frames for containers, packaging, transparency and occlusion, detailed in Appendix. To give an example, first image in Fig 3 shows our 'container' rule where other inactive objects in the cupboard are not segmented out, to show the full extent of the cupboard. Each video was annotated by one annotator to maximise temporal consistency. In addition to actively resolving annotator confusions, each video was manually verified. We created an interface for fast correction of annotations, focusing on temporal consistency.

**Splits.** In Table 2 we showcase the Train/Val/Test splits. We provide a train/val publicly-accessible split along with a held-out test set of 21 videos (≈ 20% of the frames and the masks). Note that the Test split aligns with the EPIC-KITCHENS test split, which is used for the action recognition challenge. For both Val and Test, we keep some kitchens unseen to test generality to new environments. The Train/Test annotations are available here. Evaluating on the test set is through the challenge leaderboards - see the EPIC-KITCHENS challenges for details.

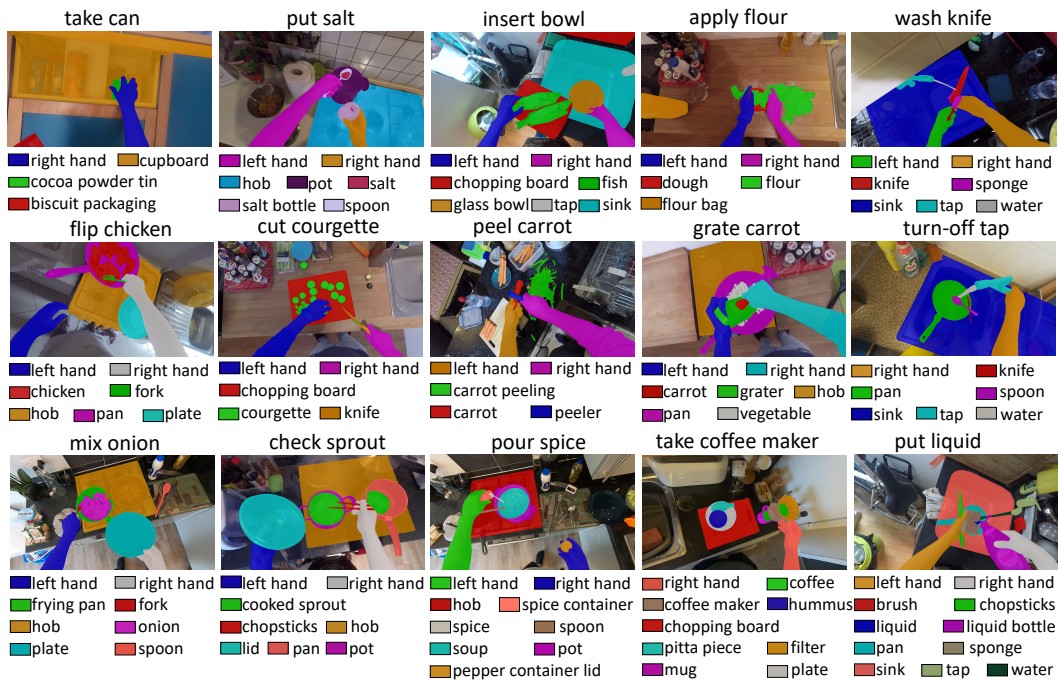

Figure 3: **Segmentations from VISOR** with entity names (legend) and ongoing action (above).

## 3.3 VISOR Object Relations and Entities

We now describe additional meta-data we collected to utilise spatial relations between the segmentations, using crowdsourcing with gold-standard checks (details in appendix).

**Entities.** EPIC-KITCHENS-100 contains 300 noun *classes*, grouping open-vocabulary entity names. Of these, VISOR segmentations cover 252 classes from the annotated 36 hours. VISOR introduces 263 new entities spanning 83 noun classes. We manually re-cluster these into the 300 EPIC-KITCHENS classes. These entities are mainly objects that are *handled* in everyday actions but not mentioned in narrations, such as 'coffee package', which is grouped into the 'package' class. We also introduce five new classes: 'left hand', 'right hand' (previously only 'hand'), 'left glove', 'right glove' (previously only 'glove') and 'foot'. As a result, VISOR contain 257 entity classes.

**Hand-Object Relations.** The activities depicted in VISOR are driven by hands interacting with the scene. We augment each hand segmentation in the dataset with contact information indicating whether it is in contact and, if so, which segment the hand is best described as contacting. We identify candidate segments as those sharing a border with the hand. Human annotators select from these segments, or alternatively identify the hand as "not in physical contact" with any object or select "none of the above" for occluded hands. Where Annotators cannot come to a consensus, we mark these as inconclusive. We ignore hands annotated with "none of the above" or inconclusive decisions during training and evaluation. Occasionally, in the kitchen, the hand is concealed by a glove, in cases of washing, cleaning or oven mats for hot surfaces. We thus also annotate all gloves to find cases where the glove is on the hand. We further annotate these on-hand gloves and their relations to active objects, in the same manner as visible hands. We consider both hands and on-hand gloves as acting hands for all relevant benchmarks.

**Exhaustive Annotations.** Recall that we annotate *active* objects rather than *all objects*. Thus, one cupboard in use can be segmented while another cupboard would not be segmented. To facilitate use of our annotations in other tasks, we flag each segment indicating whether it has been *exhaustively* annotated [17], i.e., there are no more instances of the entity in the image. Similarly, lack of consensus is marked as inconclusive. Since the task is imbalanced towards exhaustive, we add one decoy task, that is artificially made non-exhaustive, for every four real tasks to prevent annotation fatigue.

## 3.4 Dense annotations

We interpolate manual sparse annotations to produce dense interpolations where feasible. Inspired by [41], we perform forward and backward multi-mask propagation between every two manually

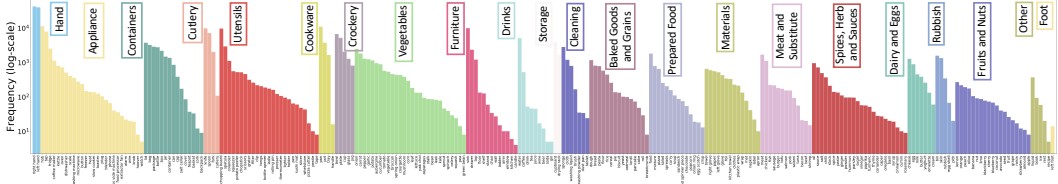

Figure 4: **Frequency of Segmented Entity Classes.** (Log y-axis) Some classes are far more frequent. Histogram is long tailed with many rare objects (e.g., 'hoover', 'cork'). Best viewed with zoom.

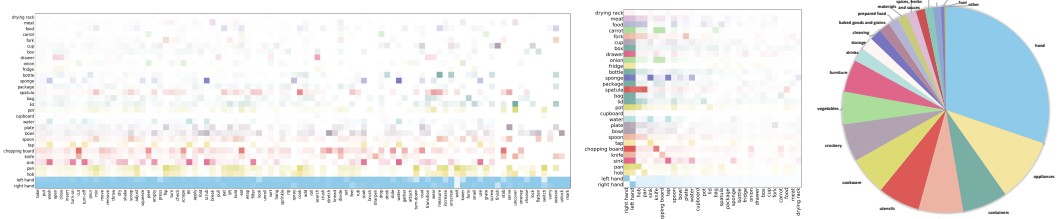

Figure 5: Frame-level co-occurences of verbs-entities (**Left**) and intra-entities (**Middle**). Percentage of annotated pixels per macro-class (**Right**) - large objects (e.g. furniture, storage) are visible.

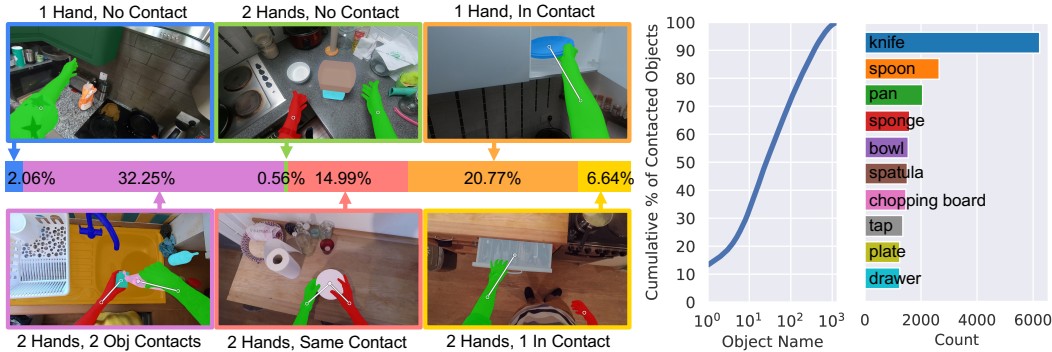

Figure 6: **Hand Object Relation Statistics.** (**Left**) Samples from 6 kinds of hand-object contacts with 1 or 2 hands. (**Middle**) Statistics about the fraction of entities making up the contacted objects, showing there is a long tail. (**Right**) Top-10 most common objects in contact with hands.

annotated *sparse* frames. The two directions' logits are averaged with a weight on each based on the distance from the manual frame. We exclude objects that appear in only one of the two frames,, except for hands which are always interpolated. Additionally, we exclude interpolations longer than 5s.

This process yields 14.5M generated masks in 3.2M train/val images (compared to 5.6M masks in 683K images in [41]). We filter interpolations by starting from the middle frame, in each interpolation, and track these automatic segmentations back to the first manually-annotated frame and forward to the last frame. We compute the average $\mathcal{J}\&\mathcal{F}$ [31] VOS metric to score each reconstructed mask. We reject all interpolated masks with $\mathcal{J}\&\mathcal{F} \leq 85$. As a result of filtering, we keep 69.4% of automatic interpolations with a total of 9.9M masks, which we release as dense annotations (sample in video).

## 4 Dataset Statistics and Analysis

Having introduced our dataset and pipeline, we present analysis by answering four questions.

**What objects are segmented?** Fig. 4 shows the segmented entity classes, clustered into 22 macro-classes, ordered by the macro-class frequency. Hands are common due to the egocentric perspective, and right hands are slightly more frequent than left ones. The next most frequent categories tend to be appliances, cutlery, and cookware like 'hob', 'knife', and 'pan'. However, the data has a long tail with segments for many rare objects such as 'watch' (5) and 'candle' (4).

**What Objects Co-Occur?** Correlations in Fig. 5 show frequent co-occurrences between actions and objects (e.g., 'measure'/'spoon') and amongst objects (e.g., 'hob'/'pan'). Although our data shows many co-occurrences, there is substantial entropy too, demonstrating the dataset's diversity.

Table 3: **VOS Performance**. We use STM [30] as a baseline and report varying training schemes. We include the scores for unseen kitchens in both Val and Test. These are considerably lower.

| Pre-train COCO [23] | Fine-Tune Davis[32]+YT[43] | Fine-Tune VISOR | Val Set Performance | | | | Test Set Performance | | | |
|---|---|---|---|---|---|---|---|---|---|---|
| | | | $\mathcal{J}\&\mathcal{F}$ | $\mathcal{J}$ | $\mathcal{F}$ | $\mathcal{J}\&\mathcal{F}_{unseen}$ | $\mathcal{J}\&\mathcal{F}$ | $\mathcal{J}$ | $\mathcal{F}$ | $\mathcal{J}\&\mathcal{F}_{unseen}$ |
| ✓ | | | 56.9 | 55.5 | 58.2 | 48.1 | 58.7 | 57.2 | 60.3 | 57.7 |
| | | ✓ | 62.8 | 60.6 | 64.9 | 53.9 | 63.7 | 61.9 | 65.5 | 62.2 |
| ✓ | ✓ | | 60.9 | 59.4 | 62.4 | 55.8 | 64.5 | 62.2 | 66.7 | 63.7 |
| ✓ | | ✓ | **75.8** | **73.6** | **78.0** | **71.2** | **78.0** | **75.8** | **81.0** | **76.6** |

**What are hands doing?** The data shows diverse hand interactions. Hands are marked as visible in 94.8% of annotated images. We annotated 72.7K hand states, of which 67.2K are in contact. Of these in-contact relations, 0.9K are contacts with on-hand gloves (76% of all gloves are marked as on-hand). Fig. 6 presents hand-object relation statistics on 77% of all images, for which conclusive annotations are present for both hands. The figure shows that 54% of these images include both hands in some contact with the world. Some objects like knives are over-represented in hand-object relations, but because the data is unscripted, there is a heavy tail: the least-contacted 90% of entities make up 24% of hand-object relations.

**How prevalent are active objects?** Although we annotate active objects, 91.1% of the objects are exhaustively annotated. 4.1% of the objects are inexhaustive, and 4.8% had inconclusive annotations. Consistently, exhaustively annotated objects include 'chopping board' and 'spatula'. In contrast, 'sponge' and 'plate' are likely to have more inactive instances in view. Inconclusive samples occur with motion blur or objects with semantic ambiguity (e.g., 'sauce', 'pan'/'pot').

## 5 Challenges and Baselines

We now define three challenges that showcase our annotations and comprise the new VISOR benchmark suite. We select the three challenges so as to demonstrate the various aspects of VISOR: short-term understanding of object transformations, in-frame hand-object relations and ultra-long video understanding respectively. Our first task (§5.1) is *Semi-Supervised Video Object Segmentation*, or tracking multiple segments from the first frame over time through a short subsequence. Our second task, detecting *Hand-Object Segmentation Relations* (§5.2), segments in-contact objects or active objects along with the manipulating hand. Our final task, titled *Where Did This Come From* (§5.3) tracks highlighted segmentations back in time to find where they were acquired from. *We report all baseline implementation details in the appendix.*

### 5.1 Semi-Supervised Video Object Segmentation (VOS)

**Definition.** We follow the definition of this task from DAVIS [32]. As explained in §3.1, we divide our videos into shorter sub-sequences. Given a sub-sequence of frames with M object masks in the first frame, the goal is to segment these through the remaining frames. Other objects not present in the first frame of the sub-sequence are excluded from this benchmark. Note that any of the $M$ objects can be occluded or out of view, and can reappear during the subsequene. We include statistics of train/val/test split for this benchmark in Appendix H.1.

**Evaluation Metrics.** Following the standard protocol used by [32, 43], we use the Jaccard Index/Intersection over Union ($\mathcal{J}$) and Boundary F-Measure ($\mathcal{F}$) metrics. Unlike other datasets, such as DAVIS [32], we use all annotated frames, including the last frame, in our evaluation. Moreover, we report the scores for unseen kitchens to assess generalisation.

**Baselines and Results.** We use Space-Time Memory Networks (STM) [30] as a baseline on this task and follow their 2-stage training strategy. First, we train using COCO [23] images by synthesising a video of 3 images from random affine transforms. We fine-tune this model using VISOR training data, again sampling 3 images from a sub-sequence. To learn long-term appearance change, we increase the gap between sampled frames during training in a curriculum learning fashion.

We report the results in Table 3. *Pre-training on COCO* suffers as many categories are specific to VISOR (e.g. left/right hands) or not present in COCO (e.g. potato), and due to the limitations of synthetic sequences. Training on *VISOR only* without pre-training gives slightly better results, but fine-tuning increases the overall score by 13% on Val. We show the gap between VISOR and [32, 43] by reporting the performance of a model trained on these benchmarks. Fig. 7 shows 5 qualitative examples.

**Code and Models.** Available from https://github.com/epic-kitchens/VISOR-VOS

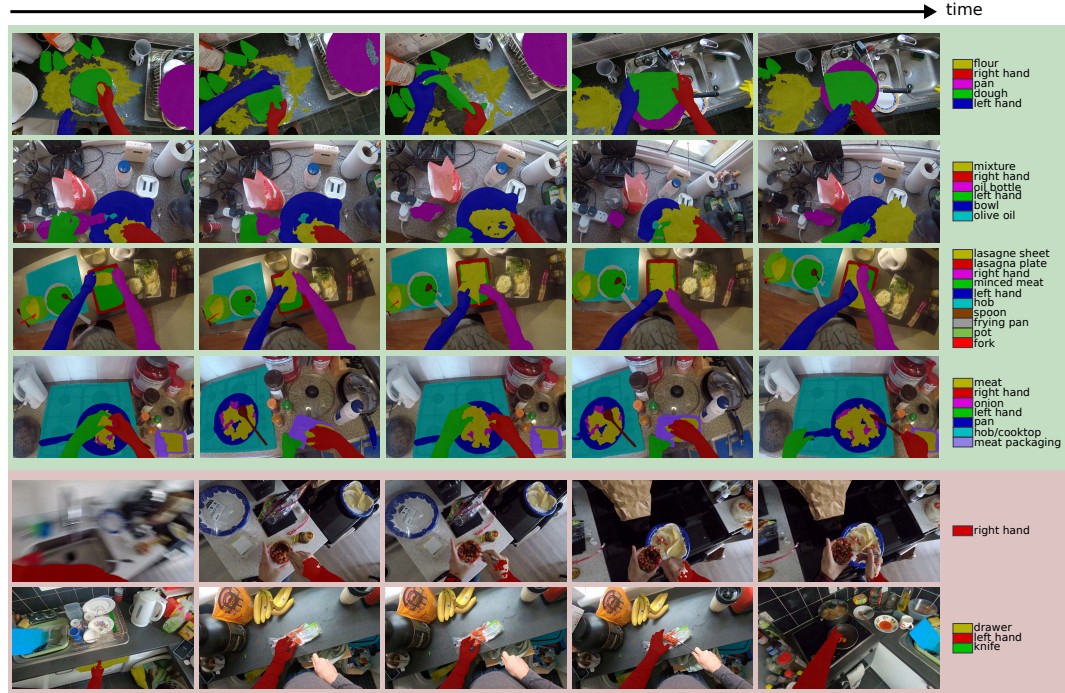

Figure 7: **Qualitative Results of STM for VOS on Validation Set.** 4 successes and 2 failures. Frames are ordered chronologically left-to-right with the first being the reference frame.

**Limitations and Challenges.** Segmenting objects through transformations offers opportunities for new research. Common failures arise from: occlusions, particularly in the reference frame (Fig. 7 row 4 where the knife is initially occluded by the drying rack), tiny objects, motion blur (Fig. 7 row 5), and drastic transformations, e.g. objects being cut, broken or mixed into ingredients.

## 5.2  Hand-Object Segmentation (HOS) Relations

**Definition.** The goal of this benchmark is to estimate the relation between the hands and the objects given a single frame as input. Our first task is *Hand-Contact-Relation* as in [35]. We characterize each hand in terms of side (left vs right), contact (contact vs no-contact), and segment each hand and contacted object. Each contacted segment must be associated with a hand, and in the case of segments with multiple connected components (e.g., multiple onions), we only use the component in contact with the hand. Our second task is *Hand-And-Active-Object*, or segmenting hands and all active objects (in-contact or otherwise). In both tasks, we also consider on-hand gloves as hands.

**Related Datasets.** Multiple past datasets have provided similar data for hands [1, 3, 27, 29] as well as hands in contact with objects [28, 34, 35]. Our work is unique in having large numbers of detailed segments for *both* hands and objects, distinguishing it from work that has boxes [35]. Most work with segments has focused only on hands [3] or has been gathered at the few thousand scale [14, 36].

**Evaluation Metrics.** We evaluate via instance segmentation tasks using the COCO Mask AP [23]. Due to large differences in AP for hands and objects, we evaluate per-category. We evaluate *Hand-Contact* by instance segmentation with in-contact objects as a class and three schemes for hand classes: all hands as one class; hands split by side; and hands split by contact state. Hand-object association is evaluated implicitly by requiring each in-contact object to associate with a hand.

**Baselines and Results.** For our baseline, we build on PointRend [21] instance segmentation. We add three auxiliary detection heads to detect hand side (left/right), contact state (contact/no-contact), and an offset to the contacted object (using the scheme and parameterisation from [35]). We select the in-contact object as the closest centre to the hand plus offset.

We use the Training set to evaluate on Val, then use both Train and Val to report results on the Test set. Table 4 shows quantitative results, and Fig. 8 shows sample qualitative examples on Val. Hands are segmented accurately, with side easier to predict than contact state. Objects are harder,

Table 4: **HOS Performance.** Hands are segmented accurately, but identifying contact is challenging.

| | | Hand-Contact | | | Hand-Active Object | |
|---|---|---|---|---|---|---|
| | Hand | Hand, Side | Hand, Contact | Object | Hand | Active Object |
| Val Mask AP | 90.9 | 87.1 | 73.5 | 30.5 | 91.1 | 24.1 |
| Test Mask AP | 95.4 | 92.4 | 78.7 | 33.7 | 95.6 | 25.7 |

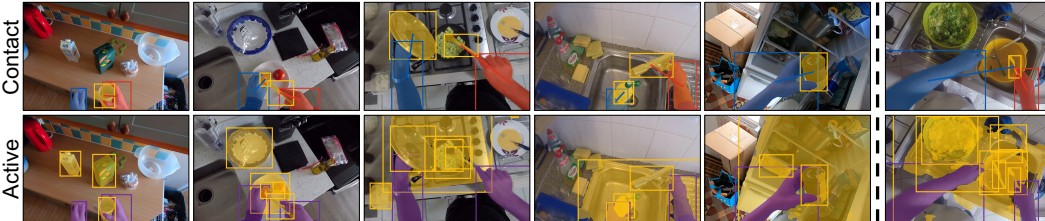

Figure 8: **HOS Relations Validation Results.** Contact (**Top**): Left Hands, Right Hands, Objects; Active Object (**Bottom**): Hands, Objects. Hands are well-segmented, as are many of the in-contact objects. In 'Active', the method finds all active objects regardless of whether they are in contact. The rightmost shows a failure when the object is not detected leading to wrong hand-object relation.

which we attribute to the long tail of objects and occlusion. Recognising active objects is harder still, particularly from single images.

**Code and Models.** Available from https://github.com/epic-kitchens/VISOR-HOS

**Limitations and Challenges.** Predicting hand contact state and segmenting contacted objects is hard due to the difficulty of distinguishing hand overlap and contact, the diversity of held objects, and hand-object and hand-hand occlusion. Using multiple frames can improve performance.

### 5.3 Where Did This Come From (WDTCF)? A Taster Challenge

**Definition.** Given a frame from an untrimmed video with a mask indicating a query object, we aim to trace the mask back through time to identify 'where did *this* come from?', where the pronoun *this* refers to the indicated mask. For tractability, we use 15 sources from which objects emerge in the dataset: {fridge, freezer, cupboard, drawer, oven, dishwasher, bag, bottle, box, package, jar, can, pan, tub, cup}. To avoid trivial solutions, the answer must be spatiotemporally grounded in an *evidence* frame(s) when the query object emerges from the source.

**Annotation Statistics.** Annotating long-term relations in videos is challenging. We obtain 222 *WDTCF* examples from 92 untrimmed videos. These form this challenge's *test* set. We encourage self-supervised learning for tackling this challenge. The distribution of the sources is long-tailed, where 'fridge' and 'cupboard' occupy the largest portions (37% and 31% respectively). Fig. 9 shows the 78 query objects and four samples with timestamp differences between the query and evidence frames. The gap between the query and evidence frames adds challenge. The gap is 5.4 mins (19K frames) on average, but it varies widely with a standard deviation of 8 mins (min=1s, max=52 mins).

When annotating, we aim to find the furthest container as a source. For example, if 'jam' emerges from a 'jar', which is a potential source container, but the 'jar' itself is seen retrieved from a 'cupboard', the correct answer to the *WDTCF* question is the 'cupboard'. In another case, where the 'jar' is only seen on the counter throughout the video, then the 'jar' would be considered as the correct source.

**Related Datasets.** This challenge is related to VQA tasks in previous datasets [9, 15, 26]. In [15], the aim is to find the previous encounter of the object itself and [9] aims to ground the object in a single image. In both, the query object is visible in the evidence image, unlike in our challenge where the query object is occluded and being manipulated. We are also looking for its source, where it was before it is first encountered, rather than when/where it was spotted last.

**Evaluation Metrics.** We evaluate by classifying the correct source (accuracy), locating the evidence frame and segmenting the source and query objects in the evidence frame (IoU). If the evidence frame is incorrect, the IoU is 0.

**Annotations and code.** Available from https://github.com/epic-kitchens/VISOR-WDTCF

**Baselines and Results.** No prior method has tackled this challenge, so we test baselines using different levels of oracle knowledge. As shown in Table 5, guessing from the source distribution

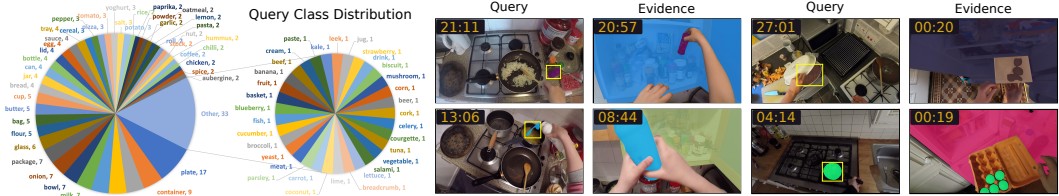

Figure 9: **WDTCF Annotations.** Distributions and examples of query entities and 4 samples.

Table 5: **WDTCF Performance.** Performance using *oracle* knowledge of distribution of source classes (prior), location of (evidence) frame, (query class) and a model trained with (GT Masks).

| Method | Prior | Evidence | Query Cls | GT Masks | Source | Query IoU | Source IoU |
|---|---|---|---|---|---|---|---|
| Random | ✓ | | | | 24.8 | - | - |
| PointRend [21] | | | ✓ | ✓ | 28.8 | 30.3 | 25.4 |
| | ✓ | | ✓ | ✓ | 33.8 | 31.6 | 29.7 |
| | | ✓ | | ✓ | 88.3 | 79.4 | 78.8 |
| | ✓ | ✓ | | ✓ | 94.1 | 79.4 | 84.1 |

gets 24.8% accuracy; while guessing the most frequent source ('fridge') achieves 37% accuracy but cannot localise the evidence frame and segment. We thus trained a PointRend instance segmentation model on the all VISOR classes using our Train and Val data to detect objects. Given the query mask, we predict its class using the mask with the highest IoU, then propose the first three frames containing both the object and a source as candidates. The relative performance suggests that the hardest task is temporally localising the evidence frame; when this is given, performance is significantly boosted.

**Limitations and Challenges.** Since objects often originate from the same source container, e.g., milk comes from the fridge, bias is expected. While we have introduced the evidence frame grounding task, shortcuts via the presence of source classes can still be used. As noted above, the baselines use oracle knowledge; an AI system to answer WDTCF is left for future work.

# 6 Conclusion and Next Steps

We introduced VISOR, a new dataset built on EPIC-KITCHENS-100 videos, with rich pixel-level annotations across space and time. Through the release of these annotations and the proposed benchmarks, we hope VISOR enables the community to investigate new problems in long-term understanding of the interplay between actions and object transformations.

# Acknowledgments and Disclosure of Funding

We gratefully acknowledge valuable support from: Michael Wray for revising the EPIC-KITCHENS-100 classes; Seung Wook Kim and Marko Boben for technical support to TORAS; Srdjan Delic for quality checks particularly on the Test set; several members of the MaVi group at Bristol for quality checking: Toby Perrett, Michael Wray, Dena Bazazian, Adriano Fragomeni, Kevin Flanagan, Daniel Whettam, Alexandros Stergiou, Jacob Chalk, Chiara Plizzari and Zhifan Zhu.

Annotations were funded by a charitable unrestricted donations to the University of Bristol from Procter and Gamble and DeepMind.

Research at the University of Bristol is supported by UKRI Engineering and Physical Sciences Research Council (EPSRC) Doctoral Training Program (DTP), EPSRC Fellowship UM-PIRE (EP/T004991/1) and EPSRC Program Grant Visual AI (EP/T028572/1). We acknowledge the use of the ESPRC funded Tier 2 facility, JADE, and the University of Bristol's Blue Crystal 4 facility.

Research at the University of Michigan is based upon work supported by the National Science Foundation under Grant No. 2006619.

Research at the University of Toronto is in part sponsored by NSERC. S.F. also acknowledges support through the Canada CIFAR AI Chair program.

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
