# OpenReview forum: "EPIC-KITCHENS VISOR Benchmark: VIdeo Segmentations and Object Relations"
_NeurIPS.cc/2022/Track/Datasets_and_Benchmarks — NeurIPS 2022 Datasets and Benchmarks _

### Official Review · Reviewer_cefW · 2022-07-22
**Novel benchmark of ego-centric videos of meal preparation**

**Rating:** 7
**Confidence:** 3
**Clarity:** The paper is clear and easy to follow.

**Strengths:**

Good paper overall.

1) A large amount of work has been obviously done, that includes developing labeling tools, creating verification procedures, formulating object categories, designing challenges, and setting up baselines.

2) The challenges are fun and inventive. Despite I have some doubts on the third one, I find the first two quite meaningful.

3) The appendix provides a detailed and comprehensive description of all procedures performed while creating the benchmark. I personally appreciate a rational and thoughtful approach to frame selection and labeling based on a common sense and supported with the statistical evidences.

4) This benchmark has a practical value: it could be used to facilitate creation of an AI-powered home assistant providing smart guidance during a meal preparation (e.g., if connected with a camera facing downwards placed underneath a cupboard).

**Weaknesses:**

1) Well, I personally dealed with the tasks implying recognising actions and objects in the kitchen. However, for a reader without such an experience, the motivation of creating this benchmark might be non-obvious. I would recommend to articulate possible usage scenarios and potential applications more clearly.

2) The test set contains 20% of the masks and 14% of the frames. Does this mean that the mask-per-frame distribution for the test set differs from the one for the remaining data? In other words, how can we be sure that the test subset is not biased w.r.t. the train subset?

3) WDTCF (Where Did This Come From) challenge seems to be fun, but a problem statement is a bit controversial to me. The fridge and jar represent different categories of containers that are not mutually exclusive and cannot be opposed in a real life (e.g., a jam might come from a jar taken out of the fridge, so various combinations are possible).

5) Among the annotated objects, there are ‘watch’ and ‘candle’. I am not really sure whether they are significant in the meal preparation context and should be annotated.

6) The VISOR annotations include such an object as 'foot'. I am terribly curious about how a foot can be used while preparing a meal (not a weakness though, but I would be grateful for an answer).

**Additional Feedback:**

See Weaknesses and Documentation.

**Correctness:**

The authors introduce three new challenges associated with the benchmark, two of which are completely new (HOS and WDTCF), so the novel evaluation protocols are designed. However, they seem quite reasonable, and rely on the standard well-known metrics (Jaccard IoU, Boundary F-measure, Mask AP).

**Documentation:**

The information about annotation process is sufficient, and the license is provided. However, I did not find any details about the dataset structure: I hope it is possible to add it to the website. The website features a leaderboard, so I assume the authors will keep the website regularly updated. This indicates an intention to attract new researchers and contribute to the field, I also hope that the dataset will remain publicly available as well.

**Ethics:**

The novel benchmark is based on EPIC-KITCHENS-100 video collection. The original videos are anonymous and signed consent of participants was provided as well, so no ethical issues seem to arise. The videos are shot by citizens of 4 countries representing 10 different nationalities, so the diversity is limited. Despite images of people not being the main focus of this benchmark, I would recommend to conduct an additional study to ensure there are no obvious racial (e.g., hands of some skin color might be recognized worse than others), gender (size of hands or the distance between the camera and hands might be an issue), or age (e.g., teens and elderly tend to have smaller hands which might be underrepresented) biases.

**Relation To Prior Work:**

The relation to EPIC-KITCHEN dataset serving as a basis for VISOR is made clear. The authors also provide a quantitative comparison with the existing datasets featuring both pixel- and action-level annotations: overall, VISOR contains longer video sequences and provides more manually annotated masks on diverse actions and objects. Having less images than the largest benchmark named YTVOS, VISOR features much more masks. VISOR does not have the largest masks-per-frame value, but it is inferior only to the UVO 0.5 dataset, which is sparse. Moreover, it is superior to all other datasets in terms of average sequence length and total capturing time.

**Summary And Contributions:**

This study is dedicated to a new semantic segmentation benchmark for segmenting and tracking hands and active objects in ego-centric videos, called VISOR. It is created on the top of EPIC-KITCHENS video dataset, yet introduces new annotations and three completely new challenges: video object segmentation (VOS), interaction understanding (Hand Object Segmentation, HOS), and long-term reasoning (Where Did This Come From? or WDTCF). The labeling is performed semi-automatically: the semantic masks are labeled via a AI-powered tool providing user guidance. Moreover, trainable STM methods are leveraged to extrapolate manual pixel-level semantic segmentation masks to the neighbor frames, giving dense video annotations.

---

### Official Review · Reviewer_LqR9 · 2022-07-23
**large scale dataset with action and pixel level annotation**

**Rating:** 7
**Confidence:** 2
**Correctness:** The construction procedure of the dat…
**Clarity:** The paper is clearly written.

**Strengths:**

The dataset is relatively larger than other datasets of the same style.
The dataset is built with proper use of AI-powered segmentation tool.
Solid and thorough inspection is performed for the dataset.

**Weaknesses:**

I don't see notable weakness in this paper.

**Additional Feedback:**

I don't have additional feedback.

**Documentation:**

Yes

**Relation To Prior Work:**

I don't have expertise in this field.

**Summary And Contributions:**

This paper extends EPIC-KITCHENS dataset for pixel level annotation.  In addition to both hands, active entity is pixel level annotated forming large scale pixel and action level annotation. The annotation was conducted on sparse frames and dense frame data is interpolated from neighboring sparse frames. This benchmark can be used for three tasks VOS, HOS and WDTCF.

---

### Official Review · Reviewer_XaV1 · 2022-07-27
**Useful dataset, richly annotated**

**Rating:** 7
**Confidence:** 4
**Correctness:** Yes, the claims are well supported.
**Clarity:** Yes.  The writing is clear.

**Strengths:**

This is a strong submission.  A richly annotated video dataset is always welcome.  The annotations seem to be of very high quality.  In addition, the authors have introduced an interesting new video reasoning problem.

**Weaknesses:**

I actually don't see many weaknesses in this submission.  I have two minor comments.

1. The discussion of related work is a bit thin, in particular with [33].  It would be great to better explain and motivate why VISOR is needed, and what problems it may enable us to solve with e.g., long-term videos?  The WDTCF task is a nice demo, but it can be motivated to highlight why such a task is needed.

2. The authors have not provided any quantitative evaluation of annotation quality.  From the demos, the annotation quality seems very high. Just to make the dataset contribution more solid, I wonder if there is a way that this can be systematically calibrated (e.g. wrt with some ground truth from mocap?)

I'd be happy to increase my score if the authors may address my concerns, especially [2].

**Additional Feedback:**

None.

**Documentation:**

Yes.

**Ethics:**

No concerns.

**Relation To Prior Work:**

The discussion is a bit short, but mostly fine. However, I'd like to hear more about how the dataset differs from UVO [33].  For example, is it mostly 12s vs. 3s on the average sequence length?

**Summary And Contributions:**

This paper introduced a richly annotated dataset for video object segmentation.  Compared with previous datasets, the new VISOR dataset has more frames, more annotated masks, and more action and entity classes.  The authors also defined a new task, Where Did This Come From (WDTCF), to show that the annotated long-term videos may enable more complex reasoning problems.  I see both the dataset and the task as major contributions.

---

### Official Review · Reviewer_t9qp · 2022-07-28
**A video segmentation benchmark with kitchen scenes**

**Rating:** 7
**Confidence:** 4
**Correctness:** All claims made in the paper are corr…

**Strengths:**

1. Data labelling pipepline is described in detail. That may be beneficial for other researchers.
2. Well-known methods are used for baselines (STM and PointRend) along with standard metrics.
3. Permissive license and no ethical concerns.

**Weaknesses:**

1. While benchmark labelling, tasks and baselines are well written, motivation for creating benchmark is described poorly. Why these three challenges are selected? What kitchen applications are possible if a method shows good quality on one of the tasks?
2. Introduction compares VISOR with previous work very briefly, I believe that adding separate Related work section would benefit the paper.

**Additional Feedback:**

No additional feedback.

**Clarity:**

The paper is well written and has extensive description of all benchmark details.

**Documentation:**

Documentation is sufficient, all necessary dataset details including hosting, maintenance, etc. are described in supplementary.

**Ethics:**

No ethical concerns since persons in videos are anonymized.

**Relation To Prior Work:**

Relation to prior work is briefly discussed in introduction.

**Summary And Contributions:**

The paper proposes VISOR benchmark, a dataset with segmented kitchen videos along with three tasks: semi-supervised video object segmentation, hand-object segmentation relations, "where did this come from" back in time segment tracking. Tasks are accompanied with metrics and baseline models. Data is under permissive license CC BY-NC 4.0.

---

### Official Review · Reviewer_sj6U · 2022-07-28
**Review for Epic-Kitchens VISOR**

**Rating:** 8
**Confidence:** 3
**Correctness:** I didn't find any incorrect claim.
**Clarity:** The submission is well written and cl…

**Strengths:**

- The dataset provides annotations on transformations (e.g. flour --> dough), which is an interesting concept still to be understood well by vision models
- The dataset provides short and long temporal actions
- The amount of annotated frames is great
- The proposed set of tasks in the benchmark has both more traditional tasks and novel attempts of making vision models able to do human-like tasks (e.g. WDTCF)
- Building on top of Epic Kitchens enriches the set of annotations and contributes to broader research
- Data is easily accessible in the format provided
- Different strategies to improve annotation quality were used

In summary, the dataset is well built, accessible and definitely contributes to the broader research community by both providing more data for existing tasks and also proposing new tasks that can push the community towards solving perception.

**Weaknesses:**

- I would have liked to be provided with code to visualise annotations rather than just images in the submission
- Statistics on annotation errors would be a plus (e.g. what proportion of manually verified annotations needed correction?)
- Reporting more diversity metrics on the annotators would help

**Additional Feedback:**

No additional feedback

**Documentation:**

- The provided dataset is easily accessible with the instructions provided in the submission and the readme file.
- Benchmarks are not 100% reproducible from the given information. Code would also have been appreciated here.

**Ethics:**

- The submission seems to correctly deal with licensing and data consent
- It would be interesting to have data distribution of raters across nationality, ethnicity, gender, etc.


**Relation To Prior Work:**

Clearly discussed.

**Summary And Contributions:**

The submission introduces new pixel level and segment level semantic annotations on top of the existing EPIC-KITCHENS-100 videos. Annotations include temporal actions over short and longer time spans (e.g. flour --> dough), pixel level segmentation of active objects, relationships between hands and active objects and a new type of annotation indicating where a given object came from. Some statistics about the data are provided.

Data is annotated via a novel proposed pipeline, done in three steps: (i) identifying active objects, (ii) pixel based annotations on the selected objects, helped by an AI-powered annotation tool, and (iii) object-hand relations. These sparse annotations are densified later with the help of models and the dense annotations (not manual) are also provided.

The submission also proposes benchmarks on three tasks: video object segmentation, hand-object segmentation and relations with objects, and a novel "where did this come from" task. Model baselines are provided for the two first tasks and an oracle-based baseline is provided for the last task.

The data is released in a clear format and is easy to use.

---

### Review · Ethics_Reviewer_Fu7i · 2022-08-22

**Recommendation:** 1

**Ethics Review:**

This current iteration of this research does not pose any severe or *immediate* ethical concerns. However, there are two issues the authors should address:

1. Diversity of participants in the video content creation: This is clearly a long-term project of honing these annotation capabilities, and I understand that sometimes researchers must start with less-than-ideal datasets (small, less diverse than ideal, etc.) because of various research constraints. But, I do think this needs to be addressed as this work develops. In Response to Reviewer cefW 2/2, the authors address this lack of diversity. I agree that it would be problematic to ascribe demographic categories after the fact. However, I would encourage the authors to think about reaching the goal of diversity of participants without relying on these categories. Race and gender categories often stand in for a set of physical characteristics that are normative among people in those categories. So, for example, rather than seeking to have participants from all racial categories, you can seek to have a range of skin tones. (The authors mention that some some of the participants in the EPIC-KITCHENS 100 videos have darker skin tones.) Looking at skin tone, rather than racial category, allows researchers to ensure that that hand/arms of people from a wide variety of backgrounds can be accurately recognized. Same with gender, which can be thought of as differences of hand size and body hair. Using this approach, the researchers can avoid possible future issues (e.g., hands with dark skin or small hands with short fingers being unrecognizable) without having to gather demographic data, which could compromise participant privacy and introduce problematic and messy social categorizations into the research process (which always runs the risk of "smuggling in" bias).

2. Regarding NeurIPS's Ethics Guidelines: Potential negative social impacts #5 Develop or extend harmful forms of surveillance is a looming future ethical crisis. While this work, as it stands, is not seeking to develop surveillance capabilities, it has obvious future applications in that area. This is a step toward more effective computational tracking and labeling of complex and small-scale bodily movements. The government and law enforcement applications are clear. But, given the possible applications posed by the authors (e.g., aiding the elderly in their homes), I am perhaps more concerned about the commercial applications. Technology that can follow and recognize hand movements would yield a treasure trove of data for commercial companies that gather data through consumer devices. We know that this data is used to predict and influence users' behavior, allowing bad actors to target the most-persuadable individuals through our increasingly individualized, personalized, and compartmentalized media ecosystem. All researchers developing technology that tracks and categorizes human behavior will have to grapple with the implications of how their work will contribute to this harmful social phenomenon.

---

### Author Response · Authors · 2022-08-16
**DOI Submission of the dataset and video trailer**

Dear Readers/Reviewers,

We now have the DOI permanent version of the dataset: [https://doi.org/10.5523/bris.2v6cgv1x04ol22qp9rm9x2j6a7](https://doi.org/10.5523/bris.2v6cgv1x04ol22qp9rm9x2j6a7)

As specified in [https://www.bristol.ac.uk/staff/researchers/data/storing-and-using-research-data/](here) this server maintains the dataset, unchanged, for 20 years.

We also prepared a short-trailer on the benchmark which we made available publicly: [https://youtu.be/yGodQAbYW_E](https://youtu.be/yGodQAbYW_E)

---

### Meta-Review · Area_Chair_G6vv · 2022-09-08

**Recommendation:** Accept
**Confidence:** 4

**Metareview:**

This paper introduces new annotations on top of the EPIC-Kitchen 100 videos, which contain pixel-level instance and semantic annotations on objects. On top of the valuable instance mask annotations, such as hand-object interaction. The annotation volume presented is significantly larger than prior works in multiple statistics. Based on the added annotations, the author also propose a new task, "Where Did This Come From (WDTCF)", to demonstrate the values of added annotations for more complex semantic tasks. This is an important area to the community, and the dataset enables both studies in important classical problems (e.g. VOS) and newly proposed tasks.

All the reviewers have suggested acceptance without major concerns to the paper. Meanwhile, the authors of the paper considered the reviewers' suggestions and improved the paper to include:
1. More discussion / separate section on related works (t9qp, XaV1)
2. Visualization tools (sj6U)
3. Baseline code (sj6U)
4. Expanded discussion of the motivation for the work and proposed task (cefW)

I believe that the authors' responses and updated materials address most of the issues raised by reviewers. I think this benchmark will be a good contribution to the computer vision community.

---

### Decision · Program_Chairs · 2022-09-16

Accept